# Unveiling hidden threats: Polycyclic aromatic hydrocarbons pollution in the glacial waters of the Meili Snow Mountains in the southeastern Tibetan Plateau

Xinyu Wen[1], Huawei Zhang[2*], Hucai Zhang[3*], Guangchao Liang[4], Binbin Ren[5], Wei Peng[1], Yan Yao[1], Mengshu Zhu[6]

**1** College of Geography and Land Engineering, Yuxi Normal University, Yuxi, Yunnan, China, **2** Faculty of Geography, Yunnan Normal University, Kunming, Yunnan, China, **3** Institute for Ecological Research and Pollution Control of Plateau Lakes, School of Ecology and Environmental Science, Yunnan University, Kunming, Yunnan, China, **4** Academy of Advanced Interdisciplinary Research, Xidian University, Xi'an, Shaanxi, China, **5** College of Tourism, Geography, History and Culture, Hulunbuir University, Hailaer, Inner Mongolia, China, **6** The State Key Laboratory of Loess and Quaternary Geology, Institute of Earth Environment, Chinese Academy of Sciences, Xi'an, Shaanxi, China

\* hwzhang@ynnu.edu.cn (HZ); zhanghc@ynu.edu.cn (HZ)

## Abstract

Polycyclic aromatic hydrocarbons (PAHs) have posed considerable threats to both ecosystems and human health. To explore their characteristics and risks in temperate glacial watersheds, water samples from the Meili Snow Mountains in the southeastern Tibetan Plateau were collected and analyzed. The results revealed that the concentrations of total PAHs (∑PAHs) ranged from 406.5 to 820.9 ng·L$^{-1}$, and the mean ∑PAH level was relatively high compared to other global studies. PAHs were characterized by low–molecular–weight congeners, mainly including fluorene, and phenanthrene. Fluorene, phenanthrene, pyrene, and benzo[a]pyrene, were prevalent throughout the Meili Snow Mountains, with concentrations ranging from 164 to 425 ng·L$^{-1}$, 23.6 to 201 ng·L$^{-1}$, 17.9 to 90.1 ng·L$^{-1}$, and 50–117 ng·L$^{-1}$, respectively. PAHs originated from heterogeneous combustion sources, such as coal combustion, vehicular emissions, and biomass burning. These variations were attributed to various factors, including altitude effects, long–range atmospheric transport, and local environmental driving patterns. Specifically, ∑PAHs in runoff rivers decreased with increasing altitude, reflecting differences in local anthropogenic activities. The risk assessment indicated that PAHs posed moderate to high ecological risks and potential carcinogenic threats. This study provides valuable insights into the safety of drinking surface water resources and the structural and functional stability of ecosystem in the Meili Snow Mountains, which is significant for improving regional ecological safety and human health.

**Data availability statement:** All relevant data are within the manuscript and its Supporting information files.

**Funding:** This work was funded by the National Natural Science Foundation of China (41861015, U2202207) and the Natural Science Foundation of the Department of Education of Inner Mongolia (NJZY22285). The funders had no role in the study design, data collection and analysis, decision to publish, or preparation of the manuscript.

**Competing interests:** The authors have declared that no competing interests exist.

## Introduction

The environmental threats posed polycyclic aromatic hydrocarbons (PAHs), have attracted widespread attention due to public awareness of health problems increases. PAHs are known for their carcinogenic, teratogenic, and mutagenic effects on humans and animals [1,2]. The United States Environmental Protection Agency (US EPA) has listed 16 PAHs as priority pollutants due to their potential health hazards. These PAHs include naphthalene (Nap), acenaphthylene (Acy), acenaphthene (Ace), fluorene (Flu), phenanthrene (Phe), anthracene (Ant), fluoranthene (Fluo), pyrene (Pyr), benz[a]anthracene (BaA), chrysene (Chry), benzo[b]fluoranthene (BbF), benzo[k]fluoranthene (BkF), benzo[a]pyrene (BaP), indeno[1,2,3–cd]pyrene (IcdP), dibenz[a,h]anthracene (DahA), and benzo[g,h,i]perylene (BghiP) [3,4]. This classification is also recognized by the European Union [5].

PAHs originate from anthropogenic and natural origins, with anthropogenic sources as the dominant contributor. These anthropogenic PAHs primarily originate from the incomplete combustion of fossil fuels and organic materials, vehicle exhaust emissions, and oil spills related to exploration activities in coastal regions [6,7]. PAHs enter aquatic environment through wet–dry deposition, sewage discharge, and surface runoff [8]. Industrial advancements and the use of various fuels have led to the widespread presence of PAHs in aquatic environment. Dissolved PAHs in waters exhibit greater bioavailability and toxicity than those in adsorbed or particulate forms [9], thereby posing threats to both aquatic ecosystem and human health. Previous studies have shown increased risk due to the direct harmful effects of dissolved PAHs on living organisms [10], i.e., development toxicity, genotoxicity, oxidative stress, carcinogenicity, and endocrine disruption [11], and their connection with human cancer [12]. Therefore, assessing the health and ecological risks of PAHs in aquatic environments is crucial for protecting the health of residents and promoting safe and healthy living environment.

Semi–volatile PAHs can disperse regionally and globally through atmospheric transport in the form of gaseous or particulate matter [13]. Glaciers, snow, and rivers serve as important vectors for the deposition of PAHs from the atmosphere, potentially impacting drinking water and agricultural water supply. Therefore, it is critical to conduct study on PAHs in temperature glacial regions to quantitatively evaluate their effects on both human health and the ecological environment [14]. Studies on PAHs in glacial watersheds have predominantly focused on the Tibetan Plateau (TP) region with minimal anthropogenic activities. Li et al measured the PAH concentrations of the Qiyi, Yuzhufeng, Xiaodongkemadi, and Gurenhekou glaciers across the TP, and found PAH concentration in the Yuzhufeng glacier were the highest and low–molecular–weight (LMW) PAHs were the most prevalent, due to the low–temperature combustion of coal and biomass [13]. Liu et al indicated PAHs in glacial meltwater and downstream river water from the easter TP originated from incomplete coal combustion and coking discharge and were no obvious carcinogenic risk to human health [15]. Li et al found the PAHs in the glaciers over the TP had low biological risk [16]. As glaciers melt, the trapped PAHs are released and can enter the runoff rivers. Therefore, it is necessary to carry out study on PAHs pollution in glacier basins with significant anthropogenic activities in the Meili Snow Mountains on the southeastern TP.

The Meili Snow Mountains (98°30′–98°46′E, 28°10′–28°41′N) constitute a continuous mountain range in the southeastern TP and serve as an integral part of the Hengduan Mountains. The highest elevation is Kawagebo Peak at 6740 m above sea level. This region hosts a significant number of temperate glaciers, which are mountain glaciers known for high accumulation and melting rates. This mountain range is vital for supplying agricultural irrigation and domestic water to residents in downstream areas, making it critical for the ecological health and well–being of these communities. The Meili Snow Mountains are located near the heavily polluted Indian subcontinent characterized by a dense population and extensive industrial and agricultural activities. Additionally, there is considerable tourism activities surrounding the Meili Snow Mountains.

Zhang et al revealed that polychlorinated biphenyls contamination in the Meili Snow Mountains, predominantly from glacier melt and atmospheric transport, poses significant ecological risks but negligible carcinogenic threats to human populations [17]. However, investigation on PAHs has not yet been conducted in the Meili Snow Mountains. Therefore, in this study, 13 glacial meltwater and 5 river water samples, were collected in October 2023. The primary objectives were to: (1) elucidate the concentration and compositional characteristics of PAHs, (2) identify potential sources of PAHs, (3) assess human health and ecological risks of PAHs, and (4) investigate the possible influencing factors of PAHs. This study aimed to provide essential data and insights into potential risks of PAHs to aquatic ecological security and public health, thereby assessing ecological conditions, and improving human health in this region.

## Materials and methods

### Sample collection

Glacial river samples were collected from different river watersheds in the Meili Snow Mountains using a clean plastic bucket in October 2023. Sampling sites included the Qunatong River (gs), Pojun River (pj), Mingyong River (my), Sinong River (sn), and Yubeng River (yb) (Fig 1). Five river water samples were collected from the downstream regions of rivers originating from glacial meltwater across various altitudinal gradients. Notably, the yb and gs watersheds are tourist destinations. In total, 18 water samples were collected at different altitudes along the glacier basins. Comprehensive details are provided in S1 Table.

During sampling, we employed the clean polypropylene suits, gloves, and a pre–cleaned stainless steel shovel to prevent pollution and ensure the accuracy of subsequent laboratory measurements. In the field, water samples were filtered using 0.7 μm glass–fiber filters (Whatman International Ltd., Maidstone, England). Samples of filtered water (2 L) was stored in low–density polyethylene bottles (Thermo Scientific), in the dark at 4°C during transport to the analytical laboratory at the Beijing Institute of Geology of Nuclear Industry, where they were subsequently stored at −18°C until analysis. Before sampling, these bottles were thoroughly rinsed with ultrapure water and acetone to remove potential organic pollutants.

### Chemicals and reagents

All chemicals used for sample processing and analysis were of analytical, liquid chromatography, or pesticide residue grade, and obtained from Wako Chemical (Osaka, Japan) and Tan–Mo Technology Co., Ltd. (Jiangsu, China). A standard PAH solution (1 mL) containing the 16 priority pollutants served as the standard sample. Deionized water, with a resistivity of 18 MΩ·cm, was obtained from a Milli–Q water purification system. Florisil, with a particle size of 60–100 mesh, was activated in an oven at 130°C for 24 h.

### Sample pretreatment

The water samples were thoroughly agitated, and 1 L was precisely measured into a separatory funnel [18]. Substitute standard solution (100 μL 2 μg·mL$^{-1}$) was added to 1 L water sample and thoroughly mixed. The pH of the water samples

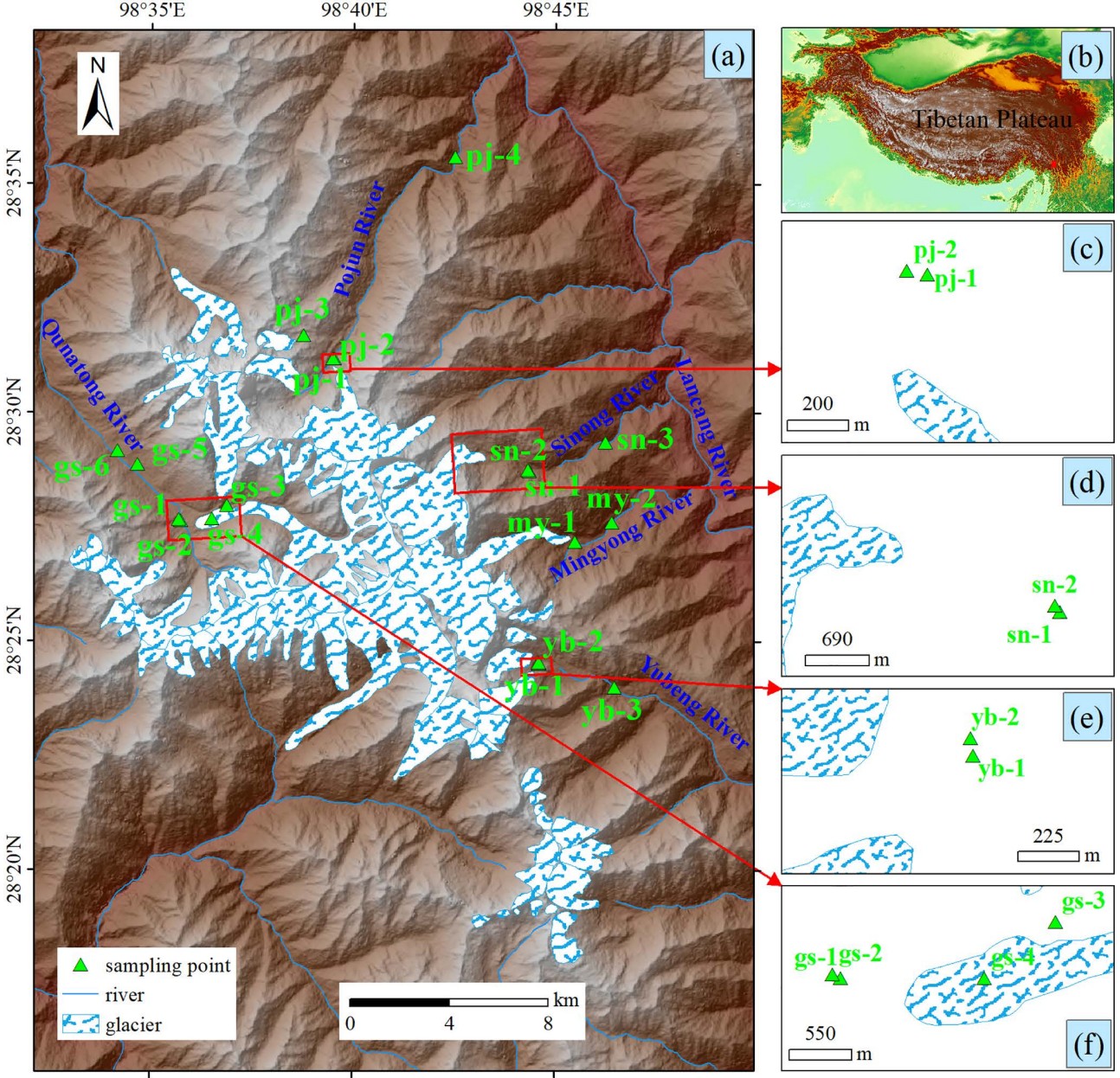

**Fig 1. Locations of the sampling points in the Meili Snow Mountains.** (a) shows the regional map of the Meili Snow Mountains, (b) indicates the location of the Meili Snow Mountains in the Tibetan Plateau, and (c), (d), (e), and (f) are partial enlarged views of glacier watersheds.Data sources: Rivers and glaciers are extracted from Landsat 8 imagery, DEM (elevation) based on Advanced Spaceborne Thermal Emission and Reflection Radiometer (ASTER), was obtained from NASA (https://www.earthdata.nasa.gov/). All sources are in the public domain and not copyrighted.

was adjusted to a range of 5–9 with hydrochloric acid. Then, 30 g sodium chloride and 50 ml n–hexane were sequentially added to the water samples, and the mixture was shaken for 5 min and allowed to stand for stratification until the aqueous phase and organic phase separate. The extraction process was repeated twice. The sample was dried using anhydrous sodium sulfate and then concentrated to 10 ml. Subsequently, the solution was concentrated to 1 mL using a nitrogen–blowing apparatus. A Florisil column was used for purification, and the eluent was further concentrated to < 1 mL. An

internal standard solution was added and adjusted to 1 mL with acetonitrile. The internal standards of PAHs consisted of naphthalene–$d_8$, acenaphthene–$d_{10}$, phenanthrene–$d_{10}$, chrysene–$d_{12}$, and pyrene–$d_{12}$. Finally, the prepared samples were stored at 4°C until analysis.

### Gas chromatography–mass spectrometry analysis

The pretreated sample was analyzed using gas chromatography–mass spectrometry (GC–MS, Clarus 600, 8547, Perkin Elmer, USA) with electron ionization [18]. Separation of target compounds was conducted using a quartz HP–5MS capillary column (30 m × 0.25 mm × 0.25 μm film thickness, Agilent Technologies) operating selected ion monitoring mode. Helium (99.99% purity) was used as the carrier gas at a constant flow rate of 1.2 mL·min$^{-1}$ during analysis. Each sample was injected in a volume of 1.0 μL in splitless injection mode. The injector temperature was maintained at 280 °C, and the electron impact ionization voltage was set to 70 eV, with the ion source temperature was set to 230 °C. The temperature program for the oven was as follows. The column was initially held at 80 °C for 2 min, subsequently increased to 180 °C at a rate of 20 °C·min$^{-1}$, and finally increased to 290 °C at a rate of 10 °C·min$^{-1}$, with a hold time of 5 min.

A five–point internal calibration curve was established for each of the 16 PAHs through serial dilution of a high–concentration stock solution (200 mg·L$^{-1}$ of 16 PAH mixture in acetonitrile) to generate five calibration at concentrations of 0.1, 0.5, 1.0, 5.0, and 10.0 μg·mL$^{-1}$. A 10 μL aliquot of each calibration standard was automatically injected into the GC–MS system using an autosampler to obtain chromatograms. Calibration curves were constructed by plotting the peak areas against corresponding concentrations, all of which exhibited excellent linearity (R$^2$ > 0.999). Quantification of individual PAHs was performed using their respective calibration curves. For MS detection, the scan mode was used a mass–to–charge (m/z) range of 35–500. Speak identification of PAHs was performed using the National Institute of Standards and Technology (NIST) mass spectral library.

### Quality assurance and quality control

The limit of detection (LoD) for the target analyte was determined using a specific method. A blank sample was subjected to injection, followed by the repeated injection zof 10 needles. Then, the standard deviation of the peak area integral at the designated retention time for the target substance was calculated, where the corresponding content was calculated with 3 standard deviation value as the detection limit. The certified material used in the analytical process consisted of a mixture of 16 PAHs in methanol (lot number: 21100429, part number: 81168a, Tan–Mo Technology Co., Ltd). The recovery rate and precision data are presented in S2 Table.

### Statistical analysis

The concentrations of PAH components at the 18 sampling sites, which were distributed across five glacial watersheds, were statistically analyzed using Origin software. Since the data were not normally distributed, the non-parametric Mann-Whitney U test (for pairwise comparisons) and Kruskal-Wallis H test (for multi-group comparisons) were applied to evaluate the significance of differences in PAHs concentrations among the different watersheds.

### Health risk assessment

Incremental lifetime cancer risk (ILCR) model established by the US EPA [19] is used to assess human health risk for infants (1–2 yrs), toddlers (5–6 yrs), children (9–12 yrs), adolescents (15–18 yrs) and adults (18–75 yrs) [20,21]. Due to the similar toxicological mechanisms of PAH congeners, toxicity equivalency factors (TEF) were used to assess toxic equivalent quotient (TEQ) relative to Bap. BaP is recognized as the most carcinogenic and mutagenic PAH, and it is the only one with adequate toxicological data to confirm its potential carcinogenicity [22]. The *TEQ* and *ILCR* formulas are as follows:

$$TEQ = \Sigma C_i \times TEF_i \tag{1}$$

$$ILCR = CSF \times TEQ \times IR \times EF \times ED \times 10^{-6}/(BW \times AT) \qquad (2)$$

where $C_i$ is the concentration of PAH congeners (ng·L$^{-1}$), and $CSF$ is the carcinogenic slope factor of BaP, quantified as 7.3 (kg·d)·mg$^{-1}$ [23]. $IR$ is water the intake rate (L·d$^{-1}$), $EF$ denotes the exposure frequency (day·yr$^{-1}$), $ED$ is the exposure duration (yr), $BW$ is the body weight (kg), $AT$ is the average time (d), and $TEF_i$ is the $TEF$ of individual PAHs (i), as shown in S3 Table. Detailed information of the primary exposure parameters is presented in S4 Table.

$ILCR$ values of less than $1 \times 10^{-6}$ indicate negligible cancer risk, while $ILCR$ values ranging from $1 \times 10^{-6}$ to $1 \times 10^{-4}$ suggest potential cancer risk. An $ILCR$ values exceeding $1 \times 10^{-4}$ indicate a high cancer risk [24,25].

### Ecological risk and potential toxicity assessment

The risk quotient ($RQ$) is used to assess the potential ecological risk of PAHs to aquatic organisms. The $RQ$ formula is as follows:

$$RQ_{NCs} = C_i/C_{NCs} \qquad (3)$$

$$RQ_{MPCs} = C_i/C_{MPCs} \qquad (4)$$

where $RQ_{NCs}$ and $RQ_{MPCs}$ is the minimum and maximum values of $RQ$, respectively. $C_i$ is the concentration of PAH congeners (ng·L$^{-1}$), while $C_{NCs}$ and $C_{MPCs}$ represent the lowest and highest risk standard values of PAHs, as detailed in S5 Table. The $RQ_{\Sigma PAHs(NCs)}$ and $RQ_{\Sigma PAHs(MPCs)}$ are the summation of 16 PAHs $RQ_{NCs}$ and $RQ_{MPCs}$, respectively, thereby providing a comprehensive assessment of 16 PAHs and accurately reflecting pollution level [26]. The ecological risk levels are presented in S6 Table.

### Backward trajectory analysis

To examine the transport mechanisms contributing to the abundance of PAHs, the HYbrid single-particle Lagrangian integrated trajectory (HYSPLIT) model was utilized to calculate the 120 h back trajectories from October 2021 to September 2023. These trajectories were generated with a termination altitude set 500 m above sea level, and cluster analysis was performed at three–month intervals.

## Results

### Concentration and occurrence of PAHs

Descriptive statistics for 16 PAHs in all samples are presented in S7 Table, while the detected PAH concentrations are shown in Fig 2. The analysis revealed that Flu, Phe, Pyr, and BaP were found at all sampling points, while Ace was detected only at pj–4 and yb–3, at the lowest concentrations, it was excluded from further analysis. The concentrations of Flu, Phe, Pyr, and BaP ranged from, 164–425 ng·L$^{-1}$, 23.6 to 168 ng·L$^{-1}$, 17.9 to 90.1 ng·L$^{-1}$, and 50–117 ng·L$^{-1}$, respectively. The relative abundance of individual PAHs followed Flu > Phe > BaP > Pyr (Fig 2a). Notably, Flu and Phe were the predominant contributors to PAH pollution. However, BaP, known for its carcinogenic properties, was detected in all samples, necessitating additional investigation. The concentrations of total PAHs (∑PAHs) ranged from 406.5 to 820.9 ng·L$^{-1}$, with a mean value of 526.9 ng·L$^{-1}$. The highest ∑PAHs appeared at yb–3 (820.9 ng·L$^{-1}$), while the lowest was at pj–1 (406.5 ng·L$^{-1}$), which suggested significant pollution in the Meili Snow Mountains. As shown in Fig 2b, the concentration of Flu showed greater variability among PAH congeners, followed by Phe, Bap, and Pyr. The significant variations in the concentration of Phe indicated a relatively more heterogeneous spatial distribution. The statistical analysis revealed significant differences ($p < 0.05$) in the concentrations of individual PAH components among the five glacial watersheds.

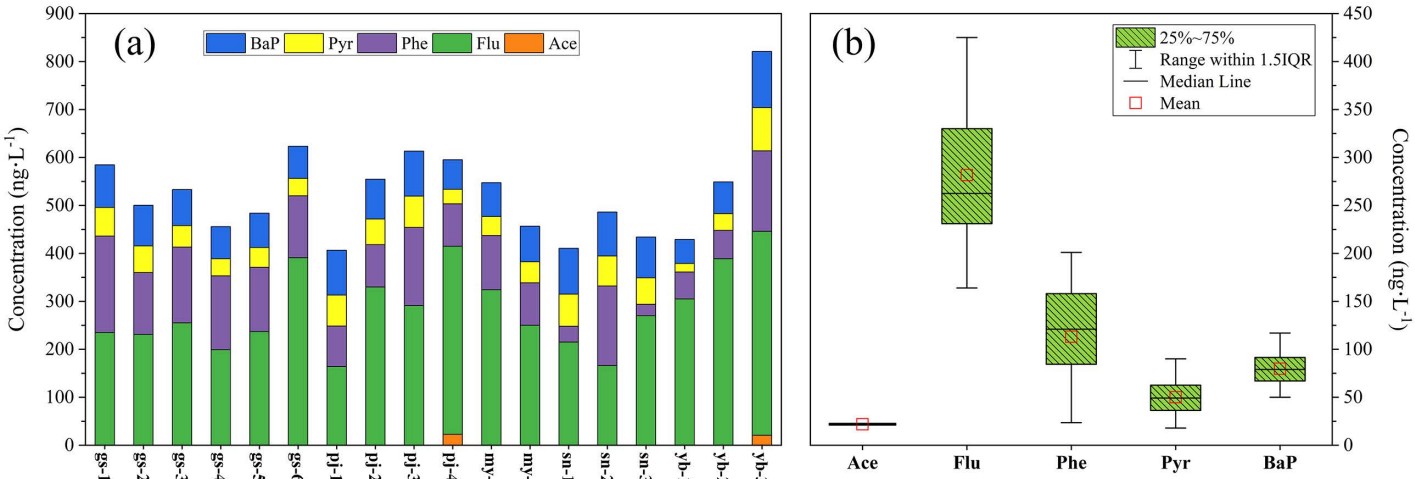

**Fig 2. Concentrations of individual PAHs at the sampling points (a) and box line plots illustrating the concentrations of Ace, Flu, Phe, Pyr, and BaP (b).**

Furthermore, a consistent spatial pattern was observed within each watershed, characterized by higher concentrations at downstream sites compared to upstream sites.

## Composition and distribution characteristics

As shown in Fig 3a, the primary compositional patterns of PAHs consisted of 3–ring (Flu, and Phe), 4–ring (Pyr), and 5–ring (BaP) in this study. The PAH compositions in all samples showed a high degree of similarity. Notably, 3–ring PAHs were the predominant pollutants, constituting 60.4–84.5% (mean value of 74.38%). This was followed by 5– and 4–ring PAHs, which accounted for 10.3–23.3% (mean value of 15.13%), and 4.2–12.9% (mean value of 9.49%), respectively. Overall, compared to high–molecular–weight (HMW, ≥ 4–ring) PAHs, LMW (2– and 3–ring) PAHs were the prevalent pollutants (Fig 3b).

Compared to meltwater samples, ∑PAHs in the river samples were relatively high, with the exception of those from the Mingyong River watershed. Analysis revealed no significant differences in the mean ∑PAHs across various glacial watersheds (S1 Fig). However, the Yubeng River and Qunatong River watersheds showed relatively high ∑PAHs, possibly due to intense anthropogenic activities associated with their status as tourist destinations. Conversely, the Sinong River watershed showed the lowest ∑PAHs.

## Health risk assessment

BaP is a potent carcinogen which serves as an indicator of PAH toxicity, as it was measured in all samples, we only employed it to assess cancer risk. The ILCR values for the five age groups are presented in Fig 4. The ILCR values varied between $0.64 \times 10^{-6}$ and $1.49 \times 10^{-6}$ (mean value of $1.01 \times 10^{-6}$) for the infant group. In the toddler group, the ILCR values ranged from $1.26 \times 10^{-6}$ to $2.95 \times 10^{-6}$ (mean value of $2.01 \times 10^{-6}$). For the child group, the ILCR values ranged from $1.92 \times 10^{-6}$ to $4.48 \times 10^{-6}$ (mean value of $3.04 \times 10^{-6}$). For the adolescent group, the ILCR values ranged from $2.22 \times 10^{-6}$ to $5.19 \times 10^{-6}$ (mean value of $3.54 \times 10^{-6}$). Finally, the adult group had ILCR values between $7.40 \times 10^{-6}$ and $17.32 \times 10^{-6}$ (mean value of $11.80 \times 10^{-6}$). Among the five age groups, the highest cancer risk was in the adult group, while the lowest risk was in the infant group. According to the current PAH levels, the cancer incidence ratios were as follows: 1.01 cases per million for infants, 2.01 cases per million for toddlers, 3.04 cases per million for children, 3.54 cases per million for

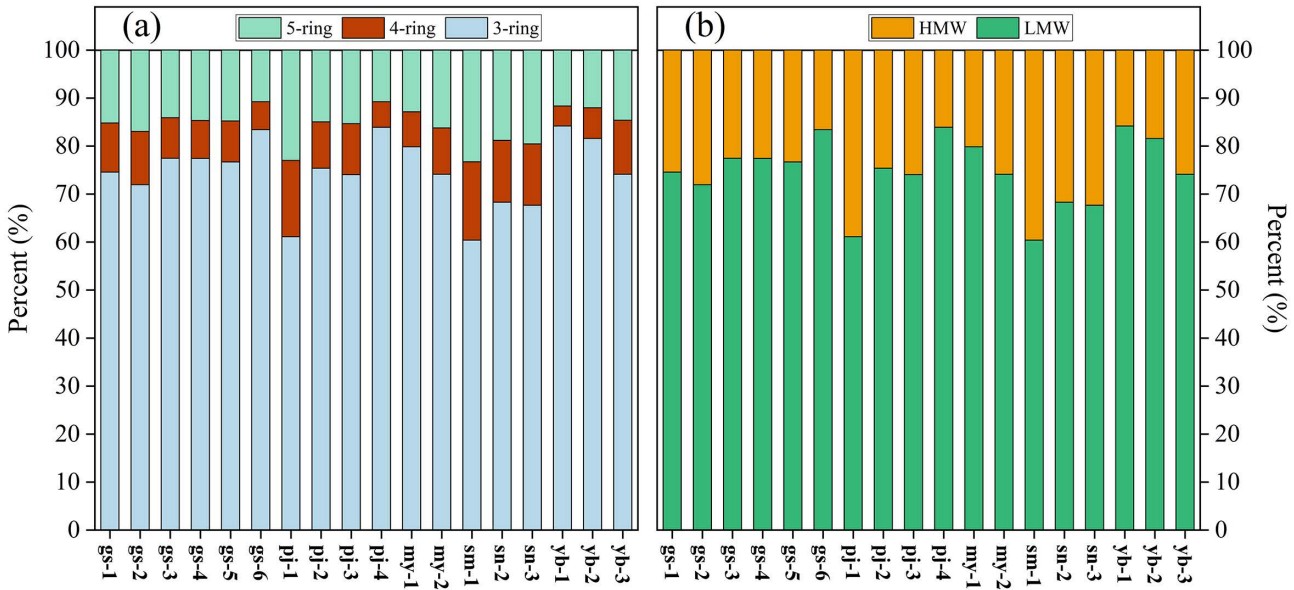

**Fig 3. Composition pattern of PAHs and contribution of LMW and HMW to ∑PAHs.**

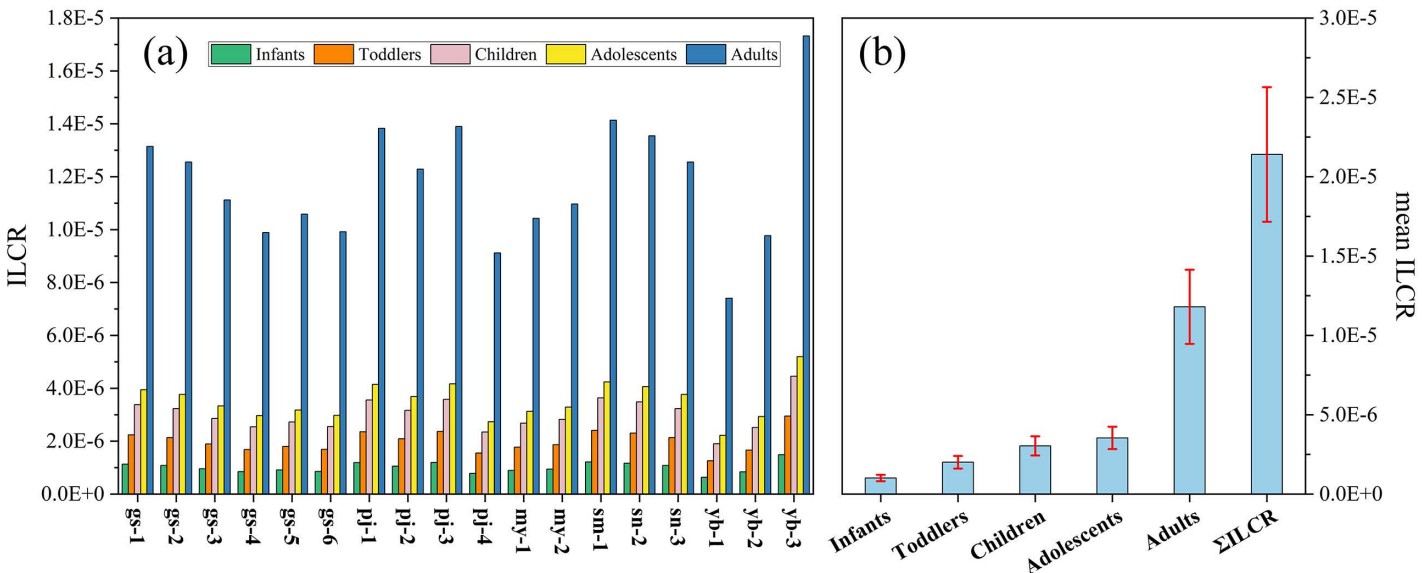

**Fig 4. ILCR values of PAHs through the ingestion of surface water exposure route: ILCR values at all sampling points (a) and mean ILCR values (b).**

adolescents, and 11.80 cases per million for adults. The lifetime cancer risk, calculated as the aggregate value of the five age groups, yielded values ranging from $13.42 \times 10^{-6}$ to $31.41 \times 10^{-6}$ (mean value of $21.40 \times 10^{-6}$). After 70 years of continuous exposure, the cancer incidence ratio was 21.40 cases per million individuals. The lowest lifetime cancer risk value was located at yb–1 ($13.42 \times 10^{-6}$) exceeding the threshold of $1 \times 10^{-6}$, demonstrating a potential cancer risk.

## Ecological risk and potential toxicity assessment

The RQ$_{NCs}$ values in all samples were more than 1 (Fig 5a), demonstrating that these PAH congeners posed moderate ecological hazards. Furthermore, the RQ$_{MPCs}$ for Flu and BaP exceeded 1 (Fig 5b), suggesting significant risk and severe toxicity to aquatic organisms. Conversely, the RQ$_{MPCs}$ values for the other PAH congeners remained below 1, with the exception of Pyr at yb–3, reflecting moderate ecological risk. Notably, Flu exhibited the highest ecological risk.

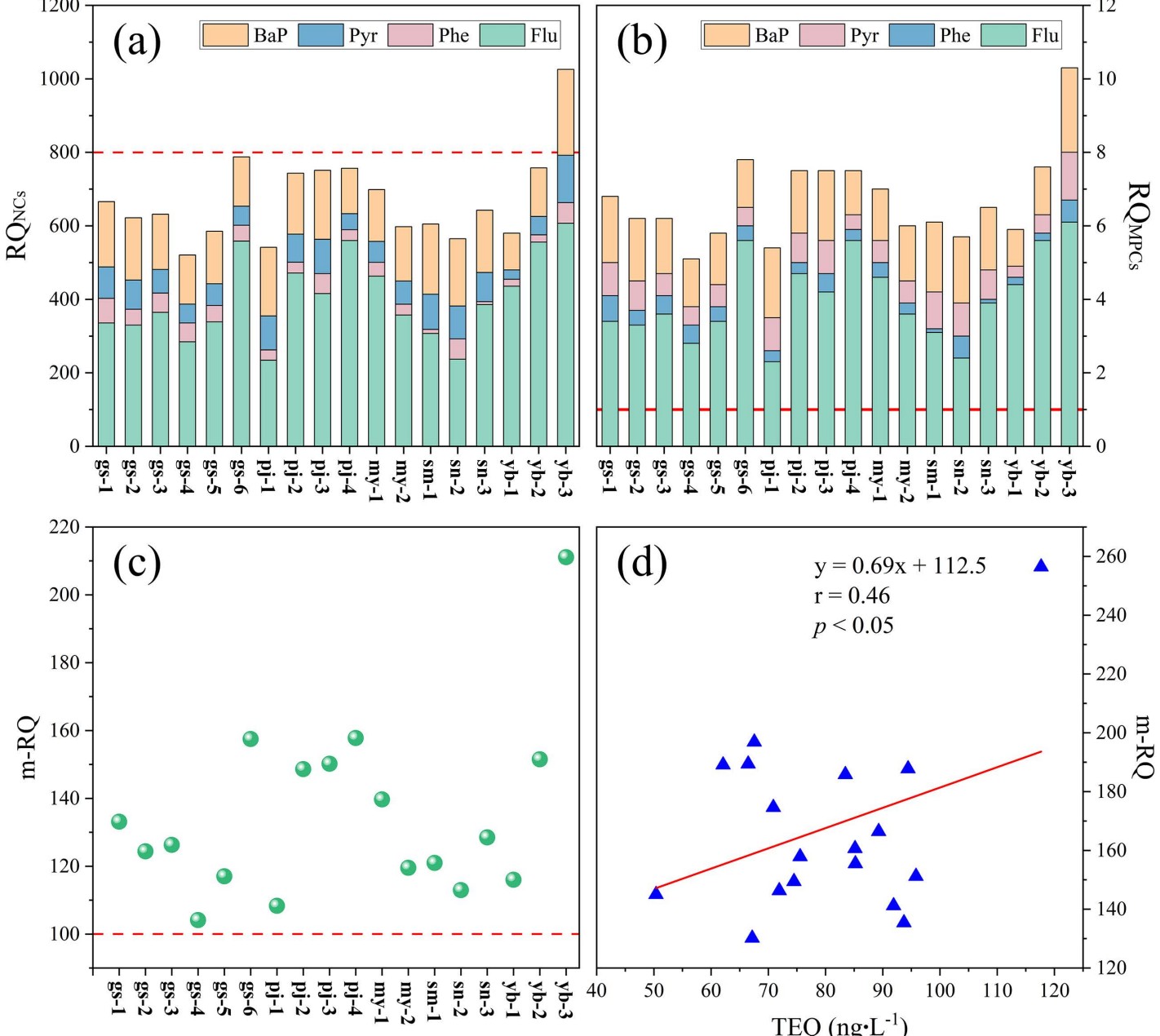

**Fig 5. Distribution of RQ and m–RQ values, and relations between m–RQ and TEQ.**

Furthermore, the RQ$_{\sum PAHs(NCs)}$ values were less than 800, with exception of yb–3 (Fig 5a), while the RQ$_{\sum PAHs(MPCs)}$ values exceeded 1 (Fig 5b). These findings suggested that the pollution levels of ΣPAHs presented a moderate–2 ecological risk; however, yb–3 was a high ecological risk. Additionally, mean–RQ (arithmetic mean of RQ$_{NCs}$) and TEQ were used to assess the potential toxicity and risk levels to aquatic organisms (Figs 5c and 5d). The m–RQ exceeded 100 at all samples, considered as high risk [27]. TEQ ranged from 50.4 to 117.7 ng·L$^{-1}$, indicating a moderate risk.

Overall, the Meili Snow Mountains exhibited a moderate to high toxicity and risk, with significant implications for ecological health. We hypothesize that this conclusion could also apply to other regions in the southeastern TP, where ∑PAHs could be an order of magnitude higher than those found in the Meili Snow Mountains.

## Discussion

### Potential source identification

The source identification is considered essential for understanding the transport and fate of PAHs in the environment. In this study, we used compositional characteristics, molecular ratios and Pearson's correlation coefficients to identify potential sources of PAHs in the Meili Snow Mountains.

PAHs originating from anthropogenic activities, petrogenic sources, and low– to moderate–temperature combustion processes were characterized by a significant presence of LWM PAHs, such as NaP, Ace, Flu, and Phe [28]. By contrast, HMW PAHs, such as Fluo, Pyr, and BaP, have been predominantly associated with pyrogenic sources and high–temperature combustion processes [29,30]. Analysis of all samples revealed the presence and concentration of Flu, Phe, Pyr, and BaP (Fig 2a), indicating that the PAHs originated from heterogeneous combustion sources. Notably, HMW PAHs were not predominant over LMW PAHs in any of the samples (Fig 3b). This suggested that contribution from pyrolytic sources, resulting from high–temperature combustion processes, was significantly lower than from low– or moderate–temperature combustion sources. Consequently, we inferred that incomplete combustion was the primary contributor to PAH emissions, which is further supported by the presence of PAH congeners indicating various sources. Phe, Flu, and Pyr were predominantly associated with the combustion of coal, biomass, and coke [9,31], while Bap was primarily released from the combustion of coal, diesel, and gasoline [32]. Furthermore, Flu and Phe were indicative of emissions from coke ovens [33].

The molecular ratios and compositional relationship are presented in Fig 6. A Flu/(Flu + Pyr) ratio exceeding 0.5 indicated diesel emissions, while below 0.5 suggested emissions from gasoline, petrol, and biomass combustion [34,35]. The

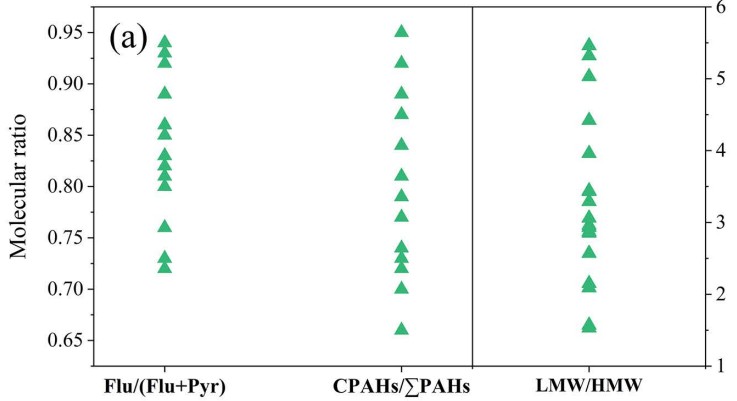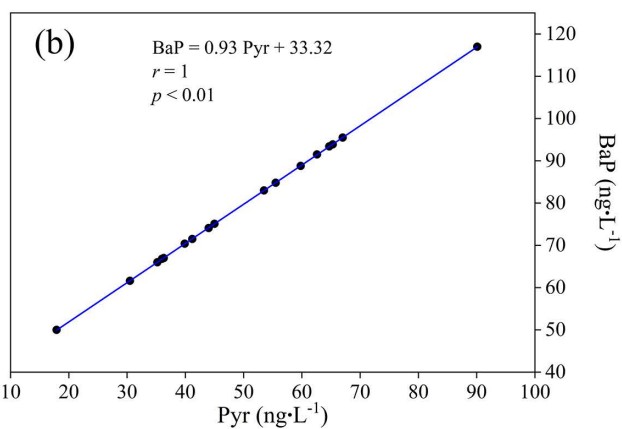

**Fig 6. Molecular ratios for selected PAHs, and compositional relationship between Pyr and BaP.**

Flu/(Flu + Pyr) ratios exceeded 0.5, indicating that diesel emissions significantly contributed to the pollution. These diesel emissions primarily originated from the transportation sector, especially from the operation of diesel vehicles and heavy trucks [36]. Furthermore, a LMW/HMW ratio of less than 1 signified combustion sources, while greater than 1 indicated petroleum sources [37]. The LMW/HMW ratios were all above 1 suggesting that pyrogenic activities were predominant, corroborating results from previous studies [13,38]. Pyrogenic sources included the incomplete combustion of organic matter, such as fossil fuels, coal, wood, and biomass, by–products from industrial processes, and emissions from gasoline or diesel–powered vehicles [39]. Additionally, nine principal non–alkylated compounds were classified as combustion PAHs (CPAHs), facilitating to differentiate between combustion and non–combustion sources [40]. A CPAH/∑PAH ratio less than 0.3 indicated petrogenic sources, while between 0.3 and 0.7 suggested a mixed source, and exceeding 0.7 implied the predominance of combustion inputs [40]. The CPAH/∑PAH ratios were above 0.7, with the exception of gs–1, gs–3, gs–4, and sn–2, further confirming the dominance of PAHs generated from combustion processes.

Relationships among PAHs due to common source were assessed by Pearson's correlation coefficients. A strong positive correlation between Pyr and BaP (Fig 6b) could be inferred from emissions of gasoline vehicles during high temperature processes.

Overall, the analysis indicated that combustion–derived PAHs, primarily originating from biomass burning, coal combustion, and vehicular emissions, were predominant in the Meili Snow Mountains.

## Comparison with previous studies

PAHs have been extensively detected in ice cores, snow, the atmosphere, and sediment in remote alpine regions [13,15,16,41–43]. Numerous studies on PAHs have been conducted in regions surrounding the TP (Fig 7a and S8 Table). Overall, the mean ∑PAH levels in river water (CR, SWGR, GR, BDR, DXR, and YTR) were higher than glacial meltwater (UPGR, BES, DG, and HG), ice cores (ICDG and ICRG), and snow (GRHK, XDKMD, YZF, and QG), with the exception of glacial meltwater from the Dagu glacier (DG) [15]. The DG is located near roadways, with significant anthropogenic activities [15]. Furthermore, the mean ∑PAHs in surface water from the SWGR and BDR reached 24750 ng·L$^{-1}$ and 5799.2 ng·L$^{-1}$, respectively [44,45]. These two rivers are situated in urban areas with high population density and strong anthropogenic activities.

The mean ∑PAH value in this study exceeded those reported in various other regions across China (Fig 7b and S8 Table), with the exception of riverine environments situated in northern China, such as LR, DXR, SR, and HR. Specifically, the mean ∑PAH value in LR was 2920.6 ng·L$^{-1}$, and PAHs primarily originated from a mixed source and combustion processes during flood and dry periods [46], respectively. In DXR, the PAHs were attributed to the incomplete combustion of coal, and emissions from cooking activities [15]. PAHs in the SR were linked to anthropogenic influences from urban oil pollution and incomplete combustion of coal and gas [47]. Furthermore, in NCL, LMW PAHs, such as Phe and Flu, remained the predominant PAHs [41].

The mean ∑PAH level in this study was higher than those reported in other regions worldwide, but lower than CR, SSR, OR, RN, and SWGR (Fig 7c and S8 Table). Importantly, water samples from CR, SSR, OR, RN, and SWGR passed through urban and industrial zones, and PAHs generated by anthropogenic activities were subsequently discharged into the river [48,49]. By contrast, the mean ∑PAHs in SWK, PA, and WSAB indicated that remote regions, far from direct anthropogenic activities, were affected by PAHs through long–range atmospheric transport (LRAT), contributing to their global distribution. Considering the high PAH level and the sensitive ecological conditions on TP, it remains imperative to identify the sources of these pollutants and implement targeted strategies to mitigate pollution.

## Influence factors

This study identified a limited number of PAHs, aligning with a previous study in the eastern TP [15], which could be explained by the difficulties encountered by PAH pollutants when reaching high–altitude regions. The contribution of LMW

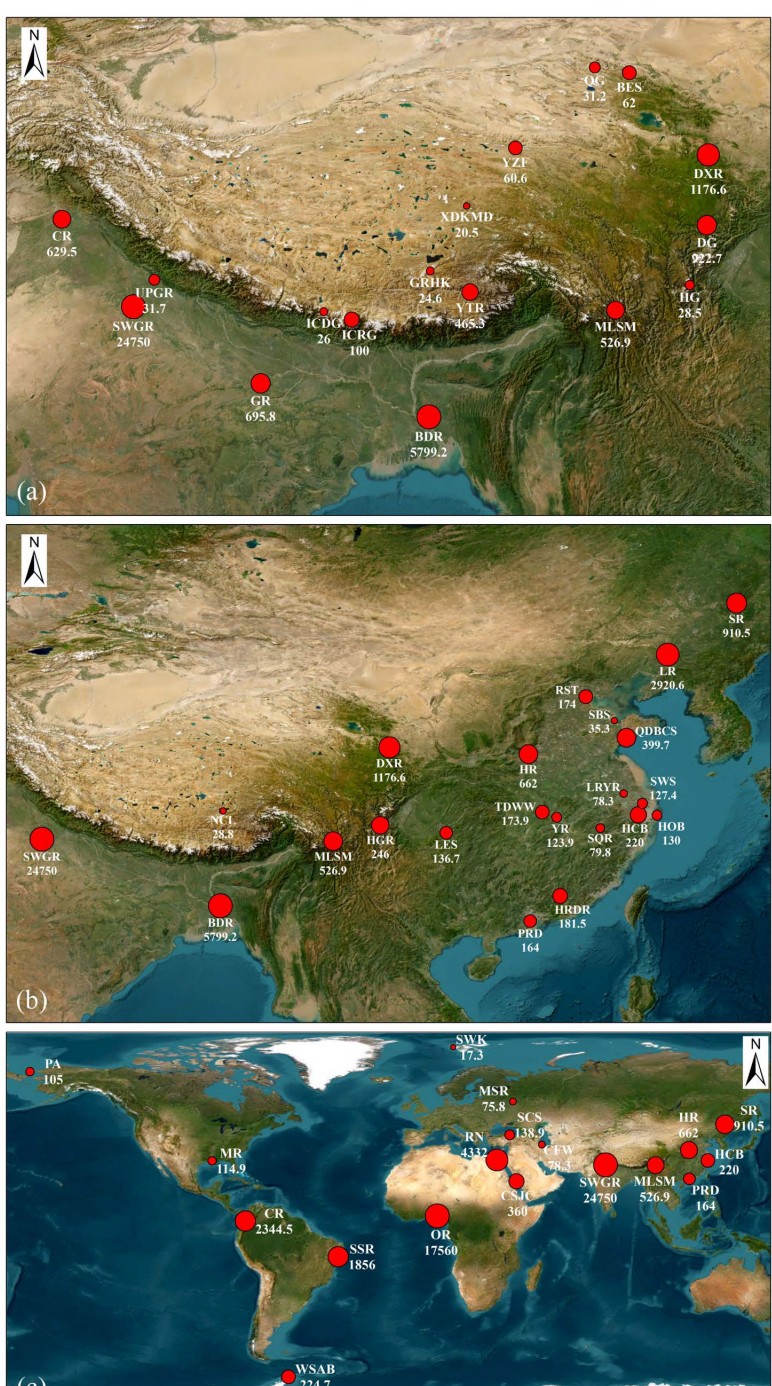

**Fig 7. Distribution of mean ∑PAHs (ng·L⁻¹) in water samples from various watersheds: areas surrounding TP (a), areas surrounding China (b), and areas around the world (c).** Detailed information is provided in S8 Table. Basemap satellite images was obtained from the World Imagery Esri layer under a CC BY license. Sources: Esri,Maxar, Earthstar Geographics, and the GIS User Community. Content is the intellectual property of Esri and is used herein with permission. Copyright © 2025 Esri and its licensors. All rights reserved.

PAHs was greater than that of HMW PAHs, due to their different solubility and volatility characteristics. Moreover, ∑PAHs were relatively high in river water located downstream of glacial confluences, likely due to the cumulative effects of PAHs in the runoff river. The results of this study were affected by regional pollution dynamics, altitude variations, and the effects of LRAT.

**Local surroundings within the various watersheds.** The compositional characteristics of PAHs in the Meili Snow Mountains were similar to those in snow and cryoconites from the TP [13,15], confirming the connection between glacial meltwater and river water [15], which could be attributed to the release of PAHs stored in snow, glaciers, and cryoconites during the melting process, subsequently entering the runoff rivers [50]. Furthermore, downstream rivers showed a cumulative effect on PAHs transported from upstream sources. Notably, the contribution of LMW PAHs was significantly greater than that of HMW PAHs (Figs 3b and 6a). This difference could be explained by the higher volatility of LMW PAHs, facilitating LRAT [51,52], whereas HMW PAHs were more likely to deposit during the condensation process [53].

Anthropogenic activities significantly increased PAH pollution levels of surface water. The Yubeng River watershed had the highest mean ∑PAH value (S1 Fig). This watershed encompasses Yupeng village, a prominent tourist destination, where emissions from vehicles, driven by an influx of visitors, contributed to high ∑PAHs. Conversely, the Sinong River watershed had the lowest mean ∑PAH value, which could be ascribed to limited tourism development, inadequate accessibility, and a sparse population. Additionally, the Qunatong River and Pojun River watersheds had relatively high mean ∑PAHs (S1 Fig), likely due to their geographical proximity and the potential LRAT.

In this study, with the exception of my–2, variations in ∑PAHs showed a significant cumulative effect on downstream river water (Fig 2a). Specially, my–2 is located in the middle reaches of the Mingyong River, while the water sample at my–1 was collected from a glacial–terminal lake. The high ∑PAHs at my–1 could be attributed to the accumulation of glacial meltwater over time, explaining why ∑PAHs at my–1 exceeded those at my–2.

**Altitude effects in the various watersheds.** Previous studies have revealed that various factors, including altitude, latitude, and distance from pollution sources, can affect ∑PAHs in environmental media [54,55]. Among these factors, altitude was identified as the most significant, as indicated by the observed inverse relationship between ∑PAHs and altitude in the eastern TP glacial basin [15]. A decreasing trend in ∑PAHs with increasing altitude was observed in certain watersheds, especially within the Qunatong River and Yubeng River watersheds (Fig 8), likely due to increased local confluence and enrichment [15]. Additionally, the sampling points located in low–altitude regions with high population density and developed tourism, were affected by various PAH emission sources. Consequently, PAHs originating from anthropogenic activities played a substantial role in adversely affecting the environment.

**LRAT and wet–dry deposition effects.** Previous studies have shown that PAHs originating from source regions can reach high–altitude region through LRAT [15,16] and wet–dry deposition processes [56]. Fig 9 illustrates a significant feature of the large–scale regional air mass in the Meili Snow Mountains, indicating that the Meili Snow Mountains is predominantly affected by Indian monsoon circulation originating from the Indian Ocean and the Bay of Bengal. Consequently, PAHs carried by water vapor were extensively distributed across the investigated areas through LRAT, as evidenced by the prevalence of LMW PAHs (Figs 3b and 6a), explained by LMW PAHs transported more effectively to remote regions than HMW PAHs [57].

Furthermore, contrary to the arid conditions from November to April, the rainy season from May to October significantly increases precipitation. The high concentration of LMW PAHs (Fig 3b) reflected the effective removal of PAHs from the atmosphere due to heavy rainfall. Apart from wet deposition, atmospheric PAHs transported through dry deposition might be another important source [58].

## Health risk assessment

The ILCR values in this study were greater than those reported in the eastern TP region [15], but lower than those of surface water from the Liaohe River in Northeast China [47]. These differences could be primarily attributed to the selection

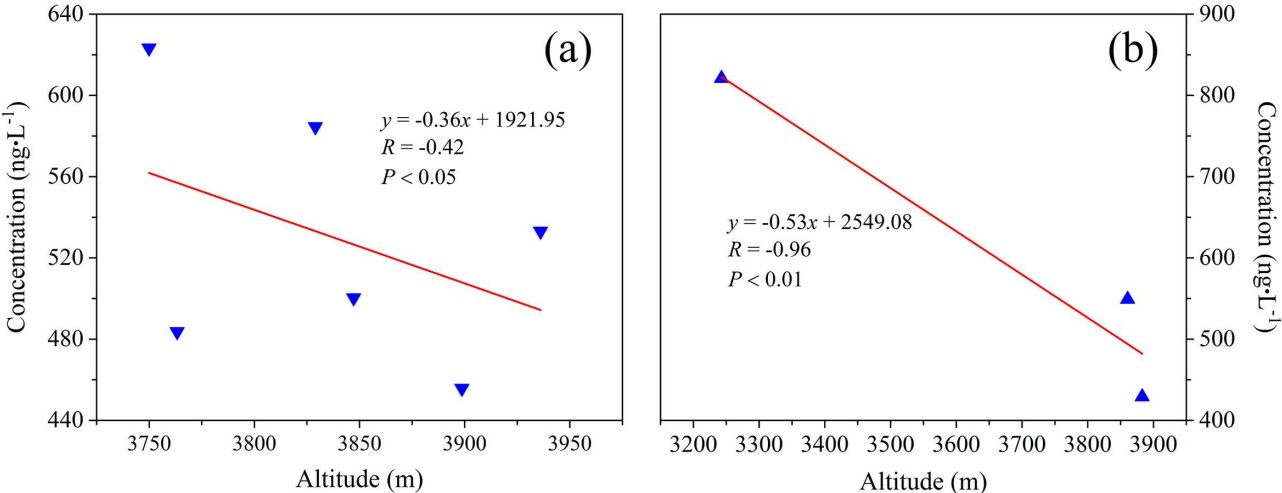

**Fig 8. Relationships between ∑PAHs and altitude: Qunatong River watershed (a), and Yubeng River watershed (b).**

of exposure parameters, and the variations in concentrations of PAH congeners. All ILCR values were within the threshold limits from $1 \times 10^{-6}$ to $1 \times 10^{-4}$, indicating that the ingestion of surface water in the investigated area posed a potential cancer risk. Importantly, infants, toddlers, and children were more vulnerable and sensitive to PAHs [59,60], increasing their risk of health complications.

PAHs can be metabolized and react as electrophilic intermediates capable of forming PAH–DNA adducts, serving as biomarkers indicative of DNA damage linked to cancer [61]. Prolonged exposure to PAHs has been shown to increase cancer risk [62]. The mean ∑ILCR values indicated that the ingestion of surface water contaminated with PAHs may account for an additional 21.40 cancer cases per million individuals (Fig 4b). This result contrasted with data reported from the eastern TP region [15], primarily due to high concentration of potent carcinogen BaP in all samples. Benzo (a) pyrene diol epoxide is a metabolite from the diol epoxide of BaP, which can react with DNA, leading to mutation and eventually cancer [63]. BaP is predominantly associated with combustion sources, including vehicular emissions, fuel oil combustions, and biomass burning [8,36,45]. This highlights the importance of implementing green mobility initiatives as promoted by national transportation systems, and converting to cleaner energy alternatives to reduce emissions resulting from traditional biomass and fuel combustion.

According to previous study [64], following the implementation of drinking water treatment plans, the mean ILCR values were $7.38 \times 10^{-7}$ for the infant group, $1.47 \times 10^{-6}$ for the toddler group, $2.22 \times 10^{-6}$ for the child group, $2.59 \times 10^{-6}$ for the adolescent group, and $8.63 \times 10^{-6}$ for the adult group. These results indicated a persistent potential cancer risk to human health.

### Ecological risk and potential toxicity assessment

The results of the ecological risk assessment (Fig 5) aligned with those reported in Aoshan Bay and Jiaozhou Bay [65]; however, they were lower than assessments conducted in cryoconties from TP glaciers [16] and seawater from Hangzhou Bay located in the western region of the East China Sea [66]. Both Flue and BaP have been shown to pose high ecological risk. Notably, yb–3, located in a tourist area, was the highest ecological risk, due to high concentration of BaP. Due to the special physical properties of HMW PAHs [67], there were difficulties in quantifying their concentrations which has ignored their ecological effects [68,69]. Compared to LMW PAHs, HMW PAHs have higher toxicity, and even at low concentration, we inferred that their ecological risk was higher. The ubiquitous presence of potent carcinogen BaP indicated

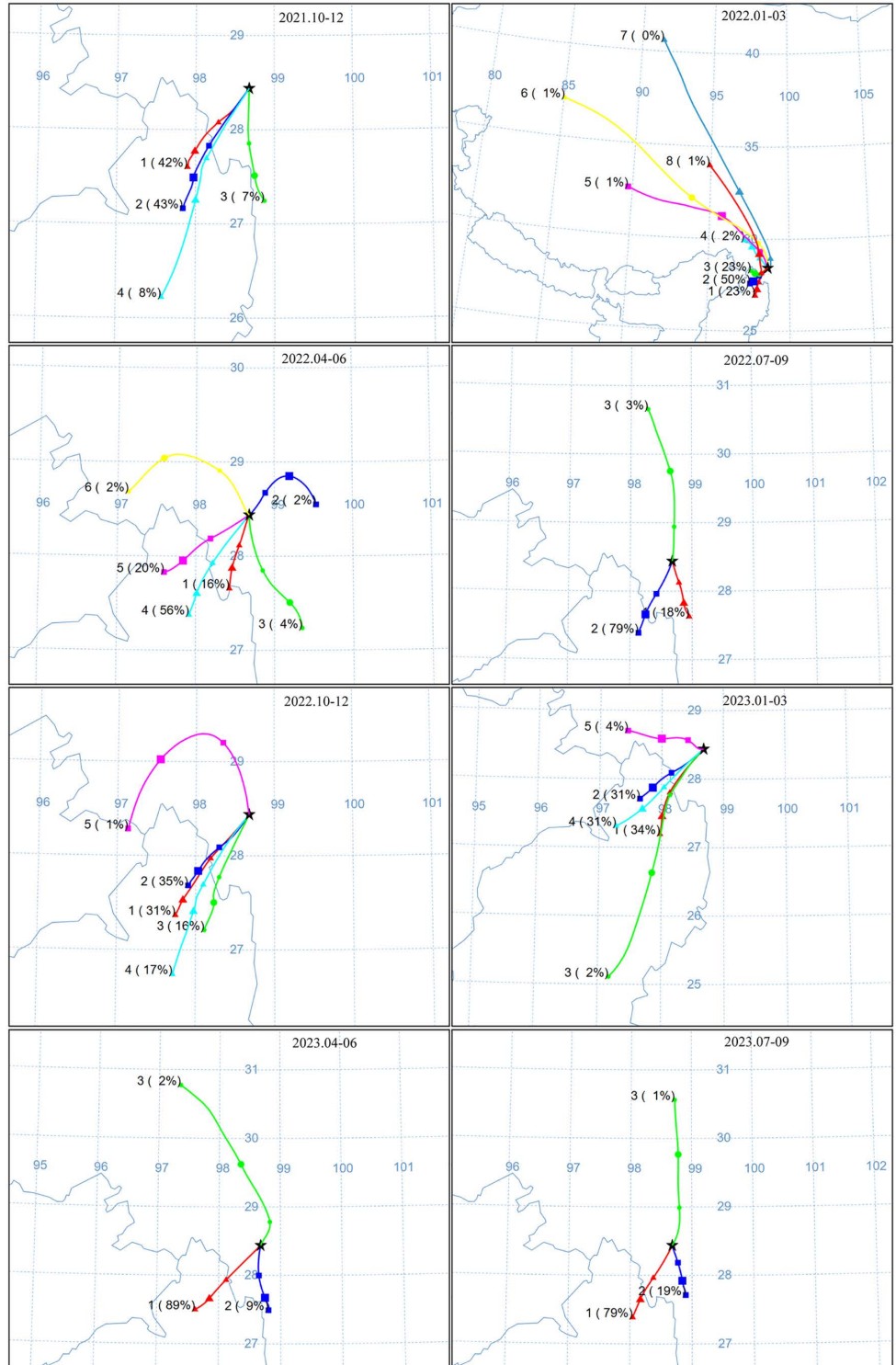

**Fig 9. Cluster analysis of 72 h back trajectories in the Meili Snow Mountains between October 2021 and October 2023.** The colored lines represent different clusters. Data sources: National border information was obtained from Natural Earth (http://www.naturalearthdata.com/). All sources are in the public domain and not copyrighted.

certain substances, typically non–toxic, exhibiting significant toxicity even at low concentrations. In the future, it will be imperative to routinely monitor the concentrations of dissolved HMW PAHs, with particular emphasis on BaP, while also identifying their primary sources. Furthermore, because PAHs in aquatic environments undergo dynamic transformations among different phases [9] and highly toxic HMW PAHs readily adsorb to particulate matter [67], subsequent studies should focus on their potential toxicity and risks to aquatic organisms of the particulate phase.

## Conclusions

In this study, we found that the ∑PAHs ranged from 406.5 to 820.9 ng·L$^{-1}$, with a mean value of 526.9 ng·L$^{-1}$. Furthermore, the PAH pollution level in this region was relatively high in comparison to other global studies. The water samples showed the presence of LMW PAH congeners, with Flu, Phe, Pyr, and BaP commonly detected. While meltwater exhibited lower ∑PAHs, downstream rivers, particularly the Qunatong River and Yubeng River watersheds, presented higher ∑PAHs. We suggested various factors, including regional pollution patterns, altitude, and LRAT, significantly affected the composition and distribution of PAHs. Importantly, we noted the mean ∑PAHs in river runoff decreased with increasing altitude, indicating that altitude–dependent variation in anthropogenic activity intensity was the primary drivers of this trend. Additionally, we observed that the PAHs originated from heterogeneous combustion, including coal combustion, vehicular emissions, and biomass burning. Backward trajectory analysis revealed that PAH pollutants from the Indian Ocean and the Bay of Bengal were transported to the Meili Snow Mountains through LRAT mechanisms, with air mass sources significantly affecting areas with elevated ∑PAHs. Furthermore, heavy rainfall had a washing effect on atmospheric PAHs, leading to the differences in influencing factors between high and low ∑PAH areas. Finally, we calculated RQ and ILCR of PAHs and found moderate to high ecological risk and potential carcinogenic threats.

This study provides a new perspective on the safety of drinking surface water resources and the structural and functional stability of ecosystem in the Meili Snow Mountains, which is of great significant for enhancing the regional ecological environment and public health. According to the study, the following interventions are proposed to mitigate localized anthropogenic emissions within scenic areas of the Meili Snow Mountains: (1) Residential Energy Transition: Implement phased replacement of firewood combustion with grid-supplied electricity for household energy needs. (2) Transportation Electrification: Facilitate conversion of gasoline-powered vehicles to zero-emission vehicles (ZEVs) in the Yubeng Scenic Area transportation fleet. These measures are projected to significantly reduce emissions of particulate matter (PM$_{2.5}$), nitrogen oxides (NO$_x$), and carbon monoxide (CO) from distributed pollution sources.

## Supporting information

**S1 Fig. Mean ∑PAHs (ng·L$^{-1}$) distribution in samples from the different watersheds.** Data sources: Rivers and glaciers are extracted from Landsat 8 imagery, DEM (elevation) based on Advanced Spaceborne Thermal Emission and Reflection Radiometer (ASTER), was obtained from NASA (https://www.earthdata.nasa.gov/). All sources are in the public domain and not copyrighted.
(DOCX)

**S1 Table. Detailed information of water samples collected from Meili Snow Mountains in the southeastern Tibetan Plateau.**
(DOCX)

**S2 Table. Details of the recovery and precision of PAH detection.**
(DOCX)

**S3 Table. TEF values of the 16 PAHs.**
(DOCX)

**S4 Table. Key exposure parameters for health risk assessment.**
(DOCX)

**S5 Table. $C_{NCs}$ and $C_{MPCs}$ values of the 16 PAHs.**
(DOCX)

**S6 Table. Risk levels of the individual PAHs and ∑PAHs.**
(DOCX)

**S7 Table. Concentrations of individual PAHs and ∑PAHs ($ng \cdot L^{-1}$) at each sampling point in the Meili Snow Mountains, southeastern Tibetan Plateau.**
(DOCX)

**S8 Table. Mean ∑PAHs ($ng \cdot L^{-1}$) in water samples from different basins around the world.**
(DOCX)

## Acknowledgments

DEM data was obtained from NASA (https://www.earthdata.nasa.gov/). Basemap satellite images was obtained from Esri,Maxar, Earthstar Geographics, and the GIS User Community. National border information was obtained from Natural Earth (http://www.naturalearthdata.com/). Thank you to the data providers mentioned above. We thank LetPub (www.letpub.com.cn) for its linguistic assistance during the preparation of this manuscript.

## Author contributions

**Conceptualization:** Huawei Zhang, Hucai Zhang.

**Data curation:** Xinyu Wen, Binbin Ren, Wei Peng, Yan Yao, Mengshu Zhu.

**Formal analysis:** Xinyu Wen, Huawei Zhang.

**Funding acquisition:** Huawei Zhang, Hucai Zhang, Binbin Ren.

**Investigation:** Xinyu Wen, Huawei Zhang.

**Methodology:** Xinyu Wen, Huawei Zhang.

**Project administration:** Huawei Zhang.

**Supervision:** Huawei Zhang.

**Writing – original draft:** Xinyu Wen, Huawei Zhang.

**Writing – review & editing:** Xinyu Wen, Huawei Zhang, Hucai Zhang, Guangchao Liang.

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
