## [Decision Letter · Decision Letter 0]

2 May 2025

Dear Dr. Zhang,

Thank you for submitting your manuscript to PLOS ONE. After careful consideration, we feel that it has merit but does not fully meet PLOS ONE’s publication criteria as it currently stands. Therefore, we invite you to submit a revised version of the manuscript that addresses the points raised during the review process.

We look forward to receiving your revised manuscript.

Kind regards,

Saidul Islam, Ph.D.

Academic Editor

PLOS ONE

Journal Requirements:

This work was funded by the National Natural Science Foundation of China (41861015, U2202207) and the Natural Science Foundation of the Department of Education of Inner Mongolia (NJZY22285). We thank LetPub (www.letpub.com.cn) for its linguistic assistance durin

This work was funded by the National Natural Science Foundation of China (41861015, U2202207) and the Natural Science Foundation of the Department of Education of Inner Mongolia (NJZY22285). The funders had no role in the study design, data collection and analysis, decision to publish, or preparation of the manuscript.

4. We note that Figures 1, 7, 9, S1 in your submission contain [map/satellite] images which may be copyrighted. All PLOS content is published under the Creative Commons Attribution License (CC BY 4.0), which means that the manuscript, images, and Supporting Information files will be freely available online, and any third party is permitted to access, download, copy, distribute, and use these materials in any way, even commercially, with proper attribution. For these reasons, we cannot publish previously copyrighted maps or satellite images created using proprietary data, such as Google software (Google Maps, Street View, and Earth). For more information, see our copyright guidelines: http://journals.plos.org/plosone/s/licenses-and-copyright.

a. You may seek permission from the original copyright holder of Figures 1, 7, 9, S1 to publish the content specifically under the CC BY 4.0 license.

Reviewers' comments:

Reviewer's Responses to Questions

**Comments to the Author**

1. Is the manuscript technically sound, and do the data support the conclusions?

Reviewer #1: Partly

Reviewer #2: Partly

2. Has the statistical analysis been performed appropriately and rigorously?

Reviewer #1: Yes

Reviewer #2: Yes

3. Have the authors made all data underlying the findings in their manuscript fully available?

Reviewer #1: No

Reviewer #2: Yes

4. Is the manuscript presented in an intelligible fashion and written in standard English?

Reviewer #1: Yes

Reviewer #2: Yes

Reviewer #1: The manuscript studies the characteristics, origin and environmental risks of PAHs contamination in glacial snowmelt and river water in the Meili Snow Mountains in the southeast of the Qinghai-Tibet Plateau known as the "Third Pole of the World." PAHs contamination levels are relatively high and are dominated by low molecular weight compounds and a variety of combustion mixing sources, which can cause moderate and higher ecological risks and potential carcinogenic threats. The results of the study provide insight into the environmental fate of organic pollutants under the dual influence of human activities and climate change. The main findings of the review are as follows:

1, the manuscript digest and the body of many clearly stated PAHs is POPs. In general, POPs are used for Persistent Organic Pollutants controlled by the Stockholm Convention, while PAHs are not included in the POPs list and it is recommended that the relevant statement be modified.

2. The state-of-the-art of POPs in glacial meltwater is not well described in the introduction. In particular, the current research status of POPs and PAHs in glacial meltwater in the Tibetan plateau is not thorough. (see Lines 71-73).

3, Lines 74-86: Does Meili Snow Mountain research exist? Only the geographical background is presented, there is no analysis of the current status of the relevant research.

Line 181:RQ PMCs typing error should be changed to RQ MPCs.

Lines 228-230: Is there any citation to suppose that tourism has a strong impact?

4、Section 4.1 Potential source identification� Flu/( Flu + Pyr) ratio and other parameters were used to source appointment, but these ratios are the rule of thumb, its use alone has some limitations, different studies in the judgment criteria also some differences, such as someone’s recommend that Flu/ (Flu + Pyr) > 0.5 indicates diesel combustion, and there are also studies proposed that Flu/ (Flu + Pyr) > 0.5 indicates biomass combustion (e.g. forest fires), coal combustion, waste incineration and other high-temperature combustion products characteristics. Therefore, when it is necessary to analyze accurate sources (e.g., differentiation of fuel types, degree of combustion) , usually using multiple molecular ratios in practice (e.g., IP/(IP + BghiP) , BaA/(BaA + Chry) , BaP/BghiP, etc.) The results are more accurate as a result of the comprehensive evaluation.

5. Conclusion: Some concrete research results are mainly described, more scattered, lack of condensation into some viewpoints and laws.

Reviewer #2: There are some issues in the manuscript which should be improved

1.Although“few investigations have been conducted on POPs in the Meili Snow Mountains”in the introduction�the research progress of other glacial areas on the Qinghai-Tibet Plateau can be added.

2.References for sample processing and analysis can be added.

3.The description of the layout of the sampling points should be added.

4.Check the format of the units throughout the text, such as ng L−1 and ng·L−1

5.The significance test of PAHs concentration differences should be added. It's too arbitrary to draw conclusions only through mean comparison.

6.Some sentences are lengthy�which can be simplified.

7.This study sampled only once and was unable to reflect seasonal changes.

8.The collection and analysis of environmental samples are lacking.

9.The description of the PAHs in the discussion can be included in the introduction.

10.Specific management suggestions should be provided according to the research.

**Do you want your identity to be public for this peer review?** For information about this choice, including consent withdrawal, please see our Privacy Policy

Reviewer #1: No

Reviewer #2: No

---

## [Author Response · Author response to Decision Letter 1]

12 Jun 2025

A point-by-point response to each comment is provided in the ‘Response to Reviewers’ document.

---

## [Decision Letter · Decision Letter 1]

15 Sep 2025

Dear Dr. Zhang,

Thank you for submitting your manuscript to PLOS ONE. After careful consideration, we feel that it has merit but does not fully meet PLOS ONE’s publication criteria as it currently stands. Therefore, we invite you to submit a revised version of the manuscript that addresses the points raised during the review process.

**ACADEMIC EDITOR:** Minor revision. Please ensure you address all reviewer comments, especially those related to statistical criteria.

We look forward to receiving your revised manuscript.

Kind regards,

Dario Rafael Olicón-Hernández

Academic Editor

PLOS ONE

Journal Requirements:

Additional Editor Comments:

Reviewer #1: The revised manuscript has largely addressed the reviewers' comments, though some modifications still require further supplementation.

1.The revised manuscript states that there has been no reported research on POPs in the Meili Snow Mountains to date. However, the authors have already published a research paper on polychlorinated biphenyls (PCBs) in the Meili Snow Mountains this year. For details, please refer to: Zhang, Huawei, et al. "Glacial Waters Under Threat: Risk Assessment and Source Identification of Polychlorinated Biphenyls in Meili Snow Mountains, Southeastern Tibetan Plateau." Toxics 13.5 (2025): 391. It is recommended to amend the relevant statements and accurately summarize the current research status.

2.Some issues were responded to but have not been fully addressed. I understand that certain facts indeed cannot be changed; however, since the questions were raised, appropriate solutions should still be provided. For example:

The claim that PAHs are POPs.

The authors insist on using average concentrations for comparative analysis without providing statistical test results, which is inappropriate.

The reviewers pointed out the lack of description regarding the collection and analysis of environmental samples, which should be supplemented. However, the authors responded by stating that the number of available samples is limited, which seems unrelated to the question raised.

Reviewer #2: All comments have been addressed

Reviewers' comments:

Reviewer's Responses to Questions

**Comments to the Author**

Reviewer #1: All comments have been addressed

Reviewer #2: All comments have been addressed

2. Is the manuscript technically sound, and do the data support the conclusions?

Reviewer #1: Yes

Reviewer #2: Yes

3. Has the statistical analysis been performed appropriately and rigorously?

Reviewer #1: No

Reviewer #2: Yes

4. Have the authors made all data underlying the findings in their manuscript fully available?

Reviewer #1: Yes

Reviewer #2: Yes

5. Is the manuscript presented in an intelligible fashion and written in standard English?

Reviewer #1: Yes

Reviewer #2: Yes

Reviewer #1: The revised manuscript has largely addressed the reviewers' comments, though some modifications still require further supplementation.

1.The revised manuscript states that there has been no reported research on POPs in the Meili Snow Mountains to date. However, the authors have already published a research paper on polychlorinated biphenyls (PCBs) in the Meili Snow Mountains this year. For details, please refer to: Zhang, Huawei, et al. "Glacial Waters Under Threat: Risk Assessment and Source Identification of Polychlorinated Biphenyls in Meili Snow Mountains, Southeastern Tibetan Plateau." Toxics 13.5 (2025): 391. It is recommended to amend the relevant statements and accurately summarize the current research status.

2.Some issues were responded to but have not been fully addressed. I understand that certain facts indeed cannot be changed; however, since the questions were raised, appropriate solutions should still be provided. For example:

The claim that PAHs are POPs.

The authors insist on using average concentrations for comparative analysis without providing statistical test results, which is inappropriate.

The reviewers pointed out the lack of description regarding the collection and analysis of environmental samples, which should be supplemented. However, the authors responded by stating that the number of available samples is limited, which seems unrelated to the question raised.

Reviewer #2: (No Response)

**Do you want your identity to be public for this peer review?** For information about this choice, including consent withdrawal, please see our Privacy Policy

Reviewer #1: No

Reviewer #2: No

---

## [Author Response · Author response to Decision Letter 2]

26 Sep 2025

We sincerely thank the editor and the reviewers for their time and constructive comments on our manuscript entitled "[Unveiling Hidden Threats: Polycyclic Aromatic Hydrocarbons Pollution in the Glacial Waters of the Meili Snow Mountains in the Southeastern Tibetan Plateau]". We have carefully considered all points raised and have revised the manuscript accordingly. We believe the manuscript has been significantly improved as a result.

Point-by-Point Response to Comments:

1. Comment: The revised manuscript states that there has been no reported research on POPs in the Meili Snow Mountains to date. However, the authors have already published a research paper on polychlorinated biphenyls (PCBs) in the Meili Snow Mountains this year. For details, please refer to: Zhang, Huawei, et al. "Glacial Waters Under Threat: Risk Assessment and Source Identification of Polychlorinated Biphenyls in Meili Snow Mountains, Southeastern Tibetan Plateau." Toxics 13.5 (2025): 391. It is recommended to amend the relevant statements and accurately summarize the current research status.

Response: We fully agree with this suggestion. We have added relevant content and reference (lines 91-93;lines 625-627) in the submitted revised manuscript.

Zhang et al revealed that polychlorinated biphenyls contamination in the Meili Snow Mountains, predominantly from glacier melt and atmospheric transport, poses significant ecological risks but negligible carcinogenic threats to human populations [17].

17. Zhang H, Yao Y, Wen X, Zhang R, Liu R. Glacial Waters Under Threat: Risk Assessment and Source Identification of Polychlorinated Biphenyls in Meili Snow Mountains, Southeastern Tibetan Plateau. Toxics. 2025; 13: 391. doi: 10.3390/toxics13050391.

2. Comment: Some issues were responded to but have not been fully addressed. I understand that certain facts indeed cannot be changed; however, since the questions were raised, appropriate solutions should still be provided. For example:

The claim that PAHs are POPs.

The authors insist on using average concentrations for comparative analysis without providing statistical test results, which is inappropriate.

The reviewers pointed out the lack of description regarding the collection and analysis of environmental samples, which should be supplemented. However, the authors responded by stating that the number of available samples is limited, which seems unrelated to the question raised.

Response: We fully agree with this suggestion.

The claim that PAHs are POPs.

We have eliminated all descriptions of PAHs as POPs in both the Abstract and the main text.

Polycyclic aromatic hydrocarbons (PAHs) have posed considerable threats to both ecosystems and human health.

The environmental threats posed polycyclic aromatic hydrocarbons (PAHs), have attracted widespread attention due to public awareness of health problems increases.

The authors insist on using average concentrations for comparative analysis without providing statistical test results, which is inappropriate.

As recommended, we have incorporated a detailed description of the statistical analysis (lines 185-190) in the Materials and Methods section and added the corresponding analytical results (lines 245-249) to the Results section.

The concentrations of PAH components at the 18 sampling sites, which were distributed across five glacial watersheds, were statistically analyzed using Origin software. Since the data were not normally distributed, the non-parametric Mann-Whitney U test (for pairwise comparisons) and Kruskal-Wallis H test (for multi-group comparisons) were applied to evaluate the significance of differences in PAHs concentrations among the different watersheds.

The statistical analysis revealed significant differences (p < 0.05) in the concentrations of individual PAH components among the five glacial watersheds. Furthermore, a consistent spatial pattern was observed within each watershed, characterized by higher concentrations at downstream sites compared to upstream sites.

The reviewers pointed out the lack of description regarding the collection and analysis of environmental samples, which should be supplemented. However, the authors responded by stating that the number of available samples is limited, which seems unrelated to the question raised.

We have revised the manuscript by making the necessary modifications and additions (lines 104-120; lines 164-174).

Glacial river samples were collected from different river watersheds in the Meili Snow Mountains using a clean plastic bucket in October 2023. Sampling sites included the Qunatong River (gs), Pojun River (pj), Mingyong River (my), Sinong River (sn), and Yubeng River (yb) (Fig. 1). Five river water samples were collected from the downstream regions of rivers originating from glacial meltwater across various altitudinal gradients. Notably, the yb and gs watersheds are tourist destinations. In total, 18 water samples were collected at different altitudes along the glacier basins. Comprehensive details are provided in S1 Table.

During sampling, we employed the clean polypropylene suits, gloves, and a pre‒cleaned stainless steel shovel to prevent pollution and ensure the accuracy of subsequent laboratory measurements. In the field, water samples were filtered using 0.7 μm glass‒fiber filters (Whatman International Ltd., Maidstone, England). Samples of filtered water (2 L) was stored in low–density polyethylene bottles (Thermo Scientific), in the dark at 4℃ during transport to the analytical laboratory at the Beijing Institute of Geology of Nuclear Industry, where they were subsequently stored at −18℃ until analysis. Before sampling, these bottles were thoroughly rinsed with ultrapure water and acetone to remove potential organic pollutants.

A five‒point internal calibration curve was established for each of the 16 PAHs through serial dilution of a high‒concentration stock solution (200 mg‧L−1 of 16 PAH mixture in acetonitrile) to generate five calibration at concentrations of 0.1, 0.5, 1.0, 5.0, and 10.0 μg‧mL−1. A 10 μL aliquot of each calibration standard was automatically injected into the GC‒MS system using an autosampler to obtain chromatograms. Calibration curves were constructed by plotting the peak areas against corresponding concentrations, all of which exhibited excellent linearity (R² > 0.999). Quantification of individual PAHs was performed using their respective calibration curves. For MS detection, the scan mode was used a mass‒to‒charge (m/z) range of 35‒500. Speak identification of PAHs was performed using the National Institute of Standards and Technology (NIST) mass spectral library.

We have made every effort to address all comments thoroughly and hope that the revisions meet with approval. Thank you again for considering our manuscript.

Sincerely,

Huawei Zhang

---

## [Editor Report · Decision Letter 2]

1 Oct 2025

Unveiling Hidden Threats: Polycyclic Aromatic Hydrocarbons Pollution in the Glacial Waters of the Meili Snow Mountains in the Southeastern Tibetan Plateau

PONE-D-25-15764R2

Dear Dr. <!--StartFragmentHuawei Zhang<!--EndFragment,

We’re pleased to inform you that your manuscript has been judged scientifically suitable for publication and will be formally accepted for publication once it meets all outstanding technical requirements.

Kind regards,

Dario Rafael Olicón-Hernández

Academic Editor

PLOS ONE

Additional Editor Comments (optional):

All the suggestions were addressed by the authors, and the work can be published in its current form
---

## [Editor Report · Acceptance letter]

PONE-D-25-15764R2

PLOS ONE

Dear Dr. Zhang,

I'm pleased to inform you that your manuscript has been deemed suitable for publication in PLOS ONE. Congratulations! Your manuscript is now being handed over to our production team.

Kind regards,

on behalf of

Dr. Dario Rafael Olicón-Hernández

Academic Editor

PLOS ONE